# REEdI Design Thinking for Developing Engineering Curricula

Fiona Boyle , Joseph Walsh, Daniel Riordan, Cathal Geary *, Padraig Kelly and Eilish Broderick

School of Science, Engineering, Technology and Mathematics, Kerry Campus, Munster Technological University, Dromtacker, Tralee, Co., V92 CX88 Kerry, Ireland; fiona.boyle@mtu.ie (F.B.); joseph.walsh@mtu.ie (J.W.); daniel.riordan@mtu.ie (D.R.); padraig.kelly@mtu.ie (P.K.); eilish.broderick@mtu.ie (E.B.)
* Correspondence: cathal.geary@mtu.ie

**Abstract:** Universities are coming under increasing pressure to re-invent the way that engineering is taught in order to produce graduates that are capable of meeting the skills needs of the country's industries. This paper described an active project where Design Thinking (DT) methodology is being applied in a novel way to Engineering Curriculum Development. Enterprise partners from a range of different manufacturing sectors participated in a series of Curriculum Development workshops and the results were cross referenced with subjects taught on existing engineering programmes internationally. This process highlighted the need for increased training in Lean, 6-Sigma, transversal and soft skills competencies, and the need to review how and when content is delivered. A survey was developed from the results of the workshops and sent out to a larger cohort of industry contacts for feedback on the proposed Engineering curriculum. Design Thinking methodology has helped ensure our customers' needs are met by building the curriculum framework around competencies identified by both industry and academia while ensuring the students engage in a significant learning experience through experiential and applied learning using the latest immersive technologies.

**Keywords:** design thinking; engineering education; innovation; education; agility

## 1. Introduction

There is significant rationale and appetite institutionally, regionally, and nationally for a new way of designing, developing and delivering engineering education in Ireland, drawing on expertise and experience internationally in the field of engineering education reform. There is an increasing argument that engineering education in Ireland is risking being a barrier to economic growth in the country [1]. Universities are coming under increasing pressure to re-invent the way that engineering is taught to students, with the ultimate aim of producing engineering graduates that are capable of meeting the skills needs of the country's industries now and into the future. A recent report by Engineers Ireland, the representative body for engineering professionals in Ireland, outlined that 91% of engineering employers identified skills shortages as a significant barrier to growth within the industry [1]. The skills shortages identified were not only in the technical areas of engineering, but also in the areas of transversal skills. Engineering employers struggle to fill roles in the mechanical and manufacturing engineering professions, which has an annual employment growth rate of 16.6% [1].

Therefore, Higher Education Institutes (HEIs) in Ireland are required to become more agile and innovative in the design, development, delivery and continuous improvement of engineering education. As part of the HEA Human Capital Initiative call for projects to enhance innovation and agility in response to future skills needs, MTU Kerry submitted a project proposal titled: Rethinking Engineering Education in Ireland (REEdI). The proposal was successful and secured funding of ca. €9 million to create a project initially focusing on the delivery of Engineering Education. The REEdI approach is to offer an agile and innovative learning programme providing personalised, flexible, and tailored options to diverse learner cohorts; from school leavers to graduating apprentices, to upskilling

industry professionals and mature students. Building on the success of world-leading cutting-edge models of engineering pedagogy, we combined an innovative method of content delivery with new immersive technologies to deliver a transformative programme of self-directed and self-scheduled learning for the next generation of industry ready engineers. The framework is truly innovative, drawing on international best practice in the field of engineering education, enabling a student-centred, project-centric and technologically innovative approach to undergraduate programme provision, equipping graduates with the skills and knowledge required to ensure they are capable of navigating the future challenges and disruptive technologies faced by the manufacturing sector in Ireland.

This paper described the approach that the REEdI project is using to develop a Bachelor of Engineering (Honors) Degree in Manufacturing and Mechanical Engineering at Munster Technological University (MTU). Furthermore, the team is utilising a variety of pedagogical aspects identified by Ruth Graham [2] as being international best-practice in engineering education reform and innovation:

1.  'Strategic industry partnerships to drive the development and continuous improvement of programmes': REEdI is addressing this challenge by having a strong consortium of manufacturing industry involved in the project.
2.  'Pathways and linkages for students to engage with the university's research activities, often building upon rigorous, applied teaching in the engineering fundamentals. REEdI is addressing this challenge by having a strong research arm to the consortium.
3.  'A wide range of technology-based extra-curricular activities and experiences available to students, many of which are student-led'. REEdI is addressing this challenge by having Virtual Reality (VR)/Augmented Reality (AR) readily available for students to use, as part of undergraduate studies.
4.  'Multiple opportunities for hands-on, experiential learning throughout the curriculum, often focusing on "problem identification as well as problem solution," and typically supported by state-of-the-art maker spaces and team working areas'. REEdI is addressing this challenge by having a state-of-the-art VR CAVE network available for students. In addition, the REEdI approach is to embed experiential learning across the curriculum.
5.  'The application of user-centered design throughout the curriculum, often linked to the development of students' entrepreneurial capabilities and/or engaged with the social responsibility agenda'. REEdI is addressing this challenge by having student-led, self-directed-learning and project-centric, work-based placement.
6.  'Emerging capabilities in online learning and blended learning; longstanding partnerships with industry that inform the engineering curriculum as well as the engineering research agenda'. REEdI is addressing this challenge by embedding online flexible, timely and relevant learning: the REEdI Topic Tree/eLearning platform. In addition, the strong research partnership will ensure research innovations are embedded into our curricula.

The REEdI initiative will provide an alternative option for engineering education and, indeed, other undergraduate disciplines across higher education in Ireland. The ultimate aim is for any HEI to be able to adopt our approach utilizing the roadmaps and framework developed and tested through REEdI. This paper aimed to demonstrate one aspect of the REEdI approach linked to points 1–6 above: how an adapted version of Design Thinking (DT) methodology can be applied in Engineering Curricula design with Industry partners from the Automotive, AgriTech, MedTech, Pharmaceutical, and general manufacturing sectors. The design model we used is an 8-stage design thinking framework (Figure 1); Steps 2 and 4 strive to make choices available, and Steps 3 and 5 involve making decisions on those choices:

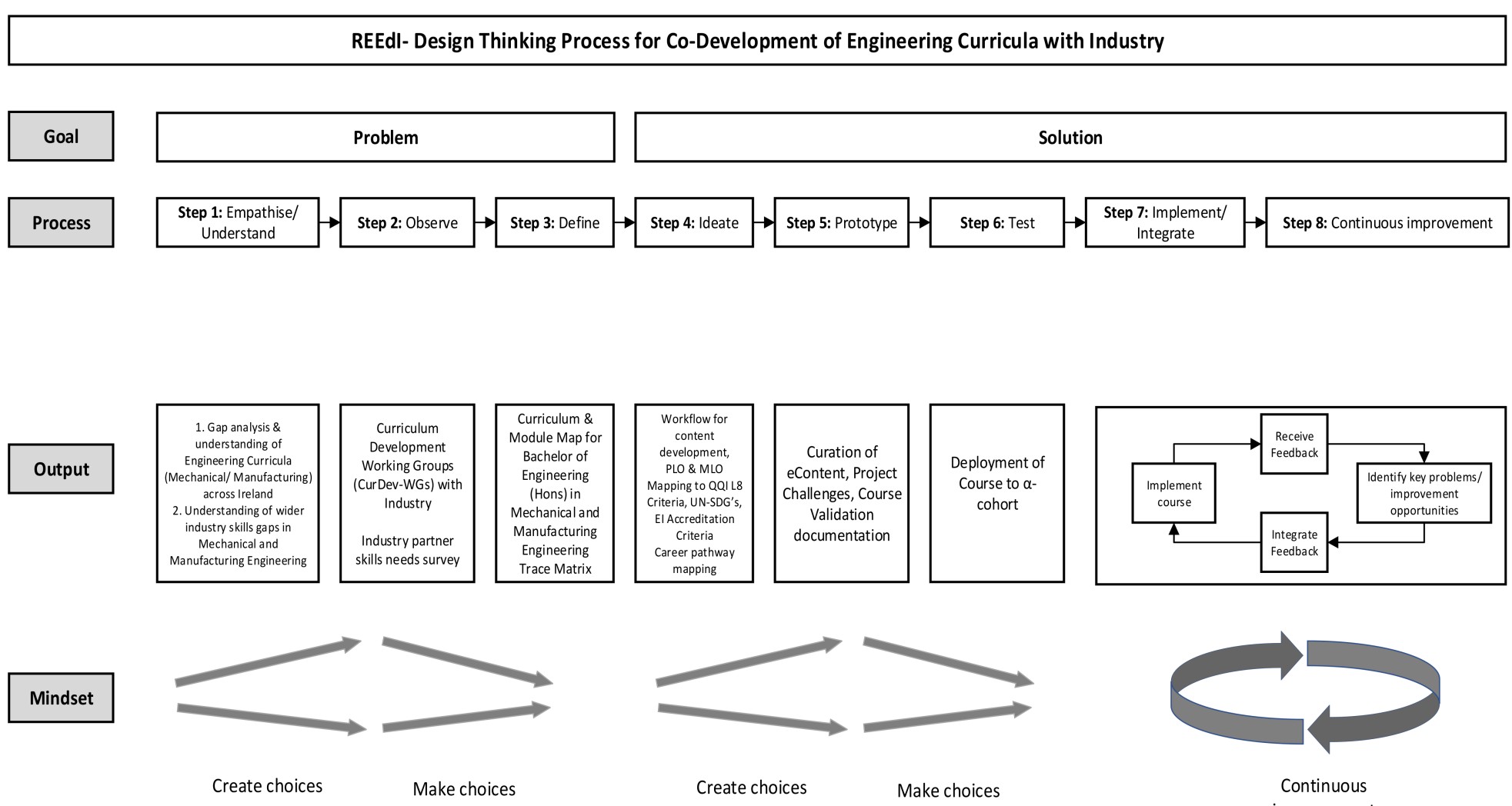

**Figure 1.** REEdI: DT Curriculum Development Process.

1.  Understand/Empathise: is where we work to understand the current situation in terms of engineering curricula content in our Technological University and nationally.
2.  Observe: as part of the co-development process with Enterprise, we established curricula development working groups (Cur-Dev-WGs) where industry experts come together to review the data generated from step 1, along with their own insights into gaps in the current offerings in undergraduate curricula. The output of these sessions is fed into step 3.
3.  Define: the data from Steps 1 and 2 enable us to define what the REEdI curriculum map will look like. The output is the REEdI curriculum and module map. In addition, a trace matrix will be developed, which will illustrate and map industry required skills and academic requirements.
4.  Ideate: during this step, outputs are the identification of programme and module learning outcomes, the modes of delivery, ideation of where immersive technologies can be utilised and implemented, how performance is assessed, and how students demonstrate competence and mastery of material.
5.  Prototype: the output of this step is the curated engineering e-content, immersive AR/VR scenarios, and performance planning resources for student engineers.
6.  Test: the output of this step is the testing of the course with the $\alpha$-cohort of student engineers.
7.  Integration: this is the integration of the course within the programmes offered at the MTU School of STEM Kerry Campus.
8.  Continuous improvement: as part of our enterprise/industry and student engineer partnership, a feedback loop was established to ensure reflection, continuous monitoring, and improvement of the REEdI Programme for future iterations and cohorts.

REEdI will transform the way we deliver undergraduate engineering education in Ireland. Building on the success of world-leading cutting-edge models of engineering pedagogy, we combined an innovative method of curricula design, content delivery utilising new immersive technologies, and student access to world-class Science Foundation Ireland research centres to deliver a transformative programme of self-directed and self-scheduled learning for the next generation of engineers. This affords a wealth of advantages in undergraduate engineering education provision. Our approach aims to be positively disruptive and transformative, with a vision to set the agenda for engineering education nationally.

## 2. Literature Review

The REEdI project is utilizing Design Thinking (DT) methodology in the design and development of Engineering Curricula. The aim is to ensure our customers' needs are met. In terms of customers, the REEdI project identifies two main customers: industry and students. Industry has a skill requirement. Student engineers embarking on a REEdI course will undertake a very different form of learning, therefore supporting them to develop a more engaged, innovative, cross disciplinary and problem-solving approach during their undergraduate degree, which is of the utmost importance.

*a.   Design Thinking: A Brief History*

Design Thinking (DT) is not a new concept. There are various definitions, principles, tools, and techniques referenced by different groups. As Schallmo et al. [3] stated, there is a clear definition and approach lacking. DT fundamentally is a human-centered, iterative approach to designing products, systems, processes, and services. While there are many approaches to DT referenced in the literature, for the purposes of REEdI and this phase of our development activity, the definition adopted was:

The approach of Design Thinking pursues the objective of developing new solutions for existing problems. These solutions are consistently oriented towards the needs of users and have a positive influence. The Design Thinking process is structured and iterative. Within the Design Thinking process, a multidisciplinary team uses techniques [3].

It is important to note that DT was first introduced in 1969 by Simon as a "way of thinking" in the design process. Thereafter, Peter Rowes expanded on DT in the methodology utilised by architects in planning [4]. In 2009, Tim Brown and Roger Martin separately further described DT in different ways, however both pointed out the application and advantages of DT in organisational change and innovation [5]. Then in 2009, Plattner and colleagues made further advancements on the methodology [6] and in 2010 the d. school at Stanford University advanced Plattner's approach [5]. This resulted in a partnership being formed between both Hasso Plattner, co-founder of the German software corporation SAP SE and the Institute of Design at Stanford [3]. Further advancements to DT methodology occurred by the global design and innovation company IDEO in 2012. The most recent advancements have been achieved by Liedtka and Ogilvie's in 2011 and Schallmo and colleagues in 2018 [3,7]. Figure 2 provides a comparison of each of the aforementioned authors' contributions to DT methodology.

Val et al. 2017 [5] outlined that all variations of DT are based around four foundation/core principles, with expansions on these outlined in Figure 2: human centred, integration oriented, doubled-diamond mindset, and prototype oriented.

- Human centred: this implies an approach that designs solutions to problems using methods and frameworks that draw on the human perspective in all steps of the process.
- Integration oriented: This involves the designer being able to simultaneously integrate and assess three factors- human factors (to make the solution meaningful), available resources (to make the solution feasible), strengths and weakness of a project (to make the solution viable). Integration oriented thinking involves abductive logic (what might be) as opposed to deductive logic (what is or should be). This ensures a balance between emotion and rationale.
- Double-Diamond mindset: This is also known as divergent and convergent thinking. Divergent meaning that choices are created and then divergent thinking meaning that decisions are then made on the options.
- Prototype oriented: This relies on the ability to do, try, fail and reiterate, constantly moving forward. This is a highly action orientated process. Engagement with end users and the customer at this point in the process is essential.

Over recent years, DT has gained traction in a variety of different fields to solve problems, foster innovation, and inspire creativity, business, engineering, education, and leadership [5,8–11].

| Approach | Principles | Procedure/ Model | Techniques |
|---|---|---|---|
| Brown *et al* 2008 (IDEO, 2012) | Integration of design thinkers, User Centricity, performing experiments, integrating external knowledge, Track projects, define budgets, build interdisciplinary team, Process run through | 3 Phase- Inspiration, Brainstorming, Implementation | Not clearly explained, however IDEO's approach includes- questionnaire structure, field research, expert interviews, storytelling, brainstorming |
| Plattner *et al* 2009 | Multidisciplinary teams, not only specialists, Consideration of different learning styles, Open creative space for teamwork, Repetition- iterations, Visualisation, brainstorming with rules, Time definition- promote spontaneity | 6 Phase- Understand, observe, point of view, ideate, prototype, test | Not explained in detail. Mentions- market research techniques, personas, storytelling, role playing, brainstorming. |
| d. School @ Stanford 2010 | Termed "Mindset" Activity, Experiment, Empathy, Visualisation, Transparency | 5 Phase- Empathise, Define, Ideate, Prototype, Test | Numerous techniques referenced. Assumption of a beginner's mindset, User camera study, Extreme users. |
| Liedtka & Ogilvie 2011 | Not clearly defined | 4 Phase- What is it? (current situation), What if? (shape the future), What wow's? (make decisions), What works? | Visualisation, journey mapping, value chain analysis, mind mapping, brainstorming, concept development, assumption testing, rapid prototyping, customer co-creation, learning launch. |
| Schallmo *et al* 2018 | What? - Human needs as a starting point, Who? - Multidisciplinary teams, How? - Iterative processes, Where? - Creative Environment | 7 Phase- Define design challenge, understand design challenge, define perspectives, gain ideas, develop prototypes, test prototypes, integrate prototypes | Not clearly defined, however, there are a significant number of activities referenced across the 7 phases. |

**Figure 2.** Comparison of Design Thinking approaches (information sourced from Schallmo et al. [3]).

*b.    Design Thinking in Education and Curriculum Development*

In the context of REEdI, a literature review was conducted to understand the application of DT in the design and development of curricula across numerous disciplines. DT was identified as a valuable approach to bolster and replace traditional methods of curriculum design and planning. Nash emphasised that traditional curriculum design approaches have heavily relied on a top-down decision attitude as opposed to a design attitude, which the author claimed has resulted in inefficient and unproductive curriculum changes [12]. Examples in the literature of DT being used in the development of new curricula and in curriculum reform include the use of DT in Medical Education [10,13], in entrepreneurship education [5,14], language courses [11,15], Tourism and Hotel Management [9], and Industrial Engineering [9]. Luka emphasised how DT in curriculum design is a novel way of ensuring 21st century skills are embedded throughout the curriculum from inception [11].

It is important to point out that the literature is laden with DT being built into engineering programmes as part of standalone courses, taught subjects, short programs, workshop scenarios, or as a pedagogical approach to embed problem-based learning throughout capstone/cornerstone projects [9,15,16]. The utilisation of DT as a taught topic and a pedagogy enables a more creative and student-centred approach for both the student and the teacher. However, there is limited literature that present the use of DT in engineering curriculum design and development [9], with no literature available (to the authors' knowledge) that utilised DT processes in the co-design and development of engineering curricula with industry. Kamp and Klassen addressed this in their vision for engineering education at TU Delft to 2030 and the authors emphasised the importance of the industry relationship in the design and development process; however, there is no process or case study presented [17]. The REEdI project aims to contribute to the literature gap in this area.

## 3. Methodology

Drawing on the literature reviewed in Section 2, the authors determined that an adaptation of the referenced DT approaches would be utilised. This eight-stage step wise process with integrated feedback loop would be suitable for our curriculum development needs (Figure 1). In terms of this paper, the REEdI-DT process presented herein is solely related to the curriculum development aspect of the project (Steps 1–3); however, the REEdI-DT process will be applied throughout our framework in the design and development of our systems, platforms, and simulations.

a.    Gap Analysis Assessment

The gap analysis or state-of-play of engineering education in Ireland assessment involved an initial review of the subjects taught on these programmes nationally (*6 HEIs) compared with sample similar engineering and technology degree courses from leading European (*2) and United States universities (*2).

b.    REEdI-DT Curriculum Development Workshop design

Virtual Curriculum Development Workshops (x4) were hosted online between June and August 2021 and were attended by 14 Industry Partners from a range of different manufacturing sectors (AgriTech, Automotive, Pharma, MedTech, General Manufacturing) based in Ireland's southwest region. The workshops were hosted by lecturers on the MTU Kerry REEdI Project (C. Geary and P. Kelly) via Zoom and utilised virtual online whiteboards (Miro®). As illustrated in Figure 3, templates were developed using virtual Post-It's on the Miro whiteboard to support systematic brainstorming and gain an in-depth understanding of industry needs as quickly as possible. The workshop series had a team consisting of:

- one Moderator,
- three Facilitators, and
- 13 industry representatives.

The moderator briefed participants on:

- the process,
- the objectives,
- the importance of a safe environment to share ideas,
- exploring ideas quickly, and
- how to share with fellow team members.

Once briefed, the workshop participants were divided into smaller discussion groups using Zoom Breakout Rooms where they discussed a question/topic posed and populated their own ideas on the Miro whiteboard: one idea per virtual post-it. After working through all the questions/topics, all participants returned to the main Zoom Workshop and one person from each Breakout Room shared their collective ideas. All virtual Post-it's were then consolidated, duplicates deleted, and then categorised or placed within the relevant discipline on the pre-prepared Miro board covering transversal, production/manufacturing, automation/robotics, and mechanical engineering/design, etc. Once captured, ideas were sorted in an iterative process to identify the key competencies required by industry and the output of this step was used to build the curriculum framework. Refer to Figure 3 for an overview of the REEdI Industry workshop series methodology.

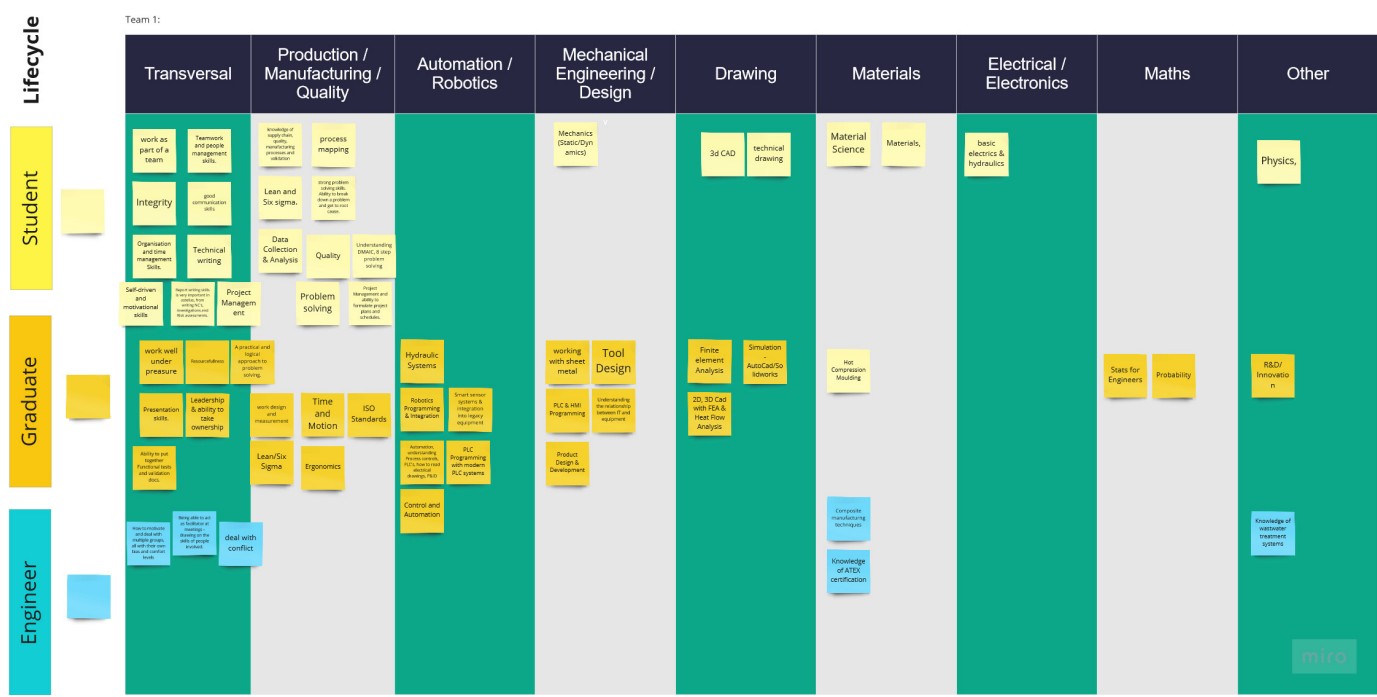

**Figure 3.** REEdI Industry Workshop: Process Flow.

Analyse: Industry partners were asked to analyse their own business requirements for Mechanical/Manufacturing Engineers under the headings of soft skills, technical skills, and academic training.

Categorise: Skills and training requirements identified were then categorised into subject areas as per the modules identified through the Gap Analysis Assessment.

Prioritise: In the final task, all skills were prioritised. The groups discussed whether they would expect a student, graduate, or engineer to be competent in each skill.

## 4. Results

This paper described an active project in engineering education reform. The results presented herein are reflective of the data available to date from steps 1 to 3 only of the REEdI-DT Curriculum Development process as illustrated in Figure 1.

Figure 4 outlines a summary of the data compiled from the Gap analysis assessments of engineering curricula nationally and internationally. Module titles were grouped into broad titles/topics as outlined in Figure 4. The topics outlined were common across all

current university courses assessed. More weighting was given to module areas that had achieved commonality across all universities reviewed.

| Topics |
|---|
| Mathematics |
| Drawing |
| Problem Solving |
| Manufacturing |
| Measurement/Metrology |
| Project Management |
| Engineering Management |
| Robotics |
| Thermodynamics/ Fluids |
| Materials |
| Project |
| Industry Placement |
| Science |
| Electrical/Electronics/Software |
| Quality |
| Maintenance/Reliability |
| Workshop/Practical |

**Figure 4.** REEdI: Summary of topics taught across Manufacturing and Mechanical Engineering degree programmes nationally and internationally.

Figure 5 outlines data gathered and categorised from the REEdI Curriculum Development workshop series with Industry partners. In general, this is described as the "next level down" of topics from Figure 4.

| Category | Transversal | Production / Manufacturing | Automation/ Robotics | Mechanical Engineering/Design | Drawing | Materials | Electronics | Mathematics | Other |
|---|---|---|---|---|---|---|---|---|---|
| **Industry required skills, knowledge, attributes** | Resilience, Stress management, assertiveness, technical writing, mentorship ability, Respect, time management, ability to work cross-functionally, problem solving mindset, communication skills, holding others accountable, creativity in problem solving, team work, integrity, people management, resourcefulness, logical thinking, presentation skills, ability to take ownership, emotional intelligence and self-awareness, humility, customer centricity, ability to deal with and overcome conflict, leadership skills, financial management / business acumen | Value stream mapping, process understanding, data visualisation, visual management, statistics for manufacturing, industry 4.0, data analytics, value engineering, IoT, Machine learning, change management, lean & six sigma, regulations/ standards, project management, product lifecycle, design control, validation, manufacturing technologies, Good manufacturing practice (GMP), packaging systems and design, overall equipment effectiveness (OEE), validation, sustainability in manufacturing process (energy, water etc) | Hydraulic systems, robotics programming & integration, understanding of automation, control of automation, PLC programming with modern PLC systems, smart sensor systems & integration into legacy equipment, automation systems architecture, understanding of programming (G-code; CNC), data analytics, control system validation | Design thinking, pneumatics, hydraulics, static & dynamic system mechanics, tool design, PLC & HMI programming, product design and development, machine design, thermodynamics, equipment selection, 3D printing, prototyping, stress/ strain testing, design of systems, water systems (WFI), metal 3D printing, equipment installation, UF/DF, basics in waste water treatment systems, vibration analysis, measurement and instrumentation, engineering metrology | Ability to read & interpret engineering drawings, symbols, tolerances drawing standards, 2D CAD, 3D CAD, AutoCAD, Solidworks, Catia, Technical Drawing | Working with different materials, material selection, material properties (including mechanical), finite element analysis (FEA), Ansys, heat flow analysis of materials, stress/ strain testing and analysis, composite manufacturing techniques, ATEX certification knowledge, heat dissipation | Basic/ fundamental electrics, electrical power calculations, P&IDs, electrical safety, electrical drawings, use of electrical equipment (e.g. multi-meter), electronic design for assembly | Applied mathematics, Data analytics & interpretation, data science, statistics for engineers, structural calculations | MS Office, Advanced Excel, health & safety, basic physics, R&D, innovation, basic understanding of VR/AR |

**Figure 5.** Overview of the outputs from the REEdI Curriculum Development Workshop series.

## 5. Discussion

There was consensus among industry partners that Transversal/Soft Skills need to be embedded key components of the REEdI Engineering Curriculum. As illustrated in Figure 6, this begins with greater self-awareness and builds towards interpersonal effectiveness while utilising a toolbox of Transversal skills. It was also highlighted that development of these Transversal/Soft Skills should begin early in the degree programme, be practised throughout on campus project challenges, and be honed during work placement projects. It was also a key finding that graduate engineers with effective communication skills are at a distinct advantage to their peers when they begin their career in industry. Furthermore, it was evident from the workshop output that, whilst a high level of competency in mathematics is widely accepted, the traditional approach is not relevant to industry engineers. Applied methods will be explored for the REEdI project.

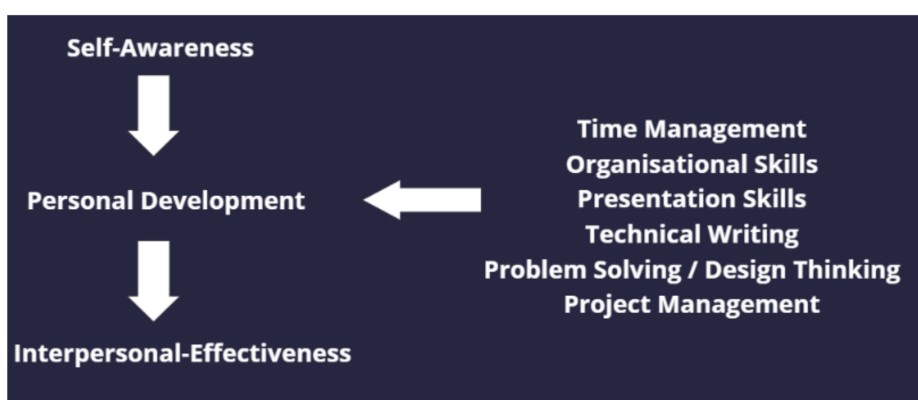

**Figure 6.** REEdI Transversal skills map.

All traditional disciplines of academic learning in Manufacturing and Mechanical Engineering roles remain necessary; however, it was evident throughout the REEdI workshops series that this needs to be developed more through experiential and applied learning. Whilst a traditional 'Just-in-Case' learning model would still apply to core Technical Skills prior to going on work placement, a 'Just-in-Time' model could be applied to more advanced skills where/when needed after this point. All aspects of Lean 6-Sigma were a recurring theme highlighted through Industry workshops as well as Data Analytics and Industry 4.0. Where possible, the Engineering Curriculum and Work Placement structure should be designed to create opportunities for experiential learning and the application of both Soft and Technical skills, e.g., Projects, Departmental Rotation on Work Placement, and Mentorship Programmes, etc. Industry partners highlighted that graduate engineers with practical experience have a much higher absorption rate of technical content, and it is believed that the REEdI project centric pillar will nurture this commonly held belief among the industry partners.

Feedback highlighted the importance of how and when the student understands the relevance of what they are learning, when they are learning it, and how or where to apply it. The workshops highlighted the opportunity to dramatically improve student engagement and their depth of understanding/retention by reviewing how and when content is delivered. In this respect, the 'gaps' identified were centred less around the content being delivered currently and more so the structure of existing Engineering Programmes. Traditional Modules are primarily structured around academic constraints. The feedback gathered through the workshop series suggested the traditional semesterised approach is driving 'Just-in-Case' Learning. The REEdI micro-module approach allows specialised content to be accessed/delivered on-demand and on a 'Just-In-Time' basis to address these nuances. Finally, the uses and effectiveness of emerging technologies such as AR/VR and their application in manufacturing are yet to be fully understood by the industry partners involved. It is important that engineers of the future have the knowledge and skills to

bring new ideas, such as AR/VR and Industry 4.0, into companies and be capable of applying them.

While this paper relates solely to the curriculum development aspect of the project (Figure 1, Steps 1–3), the Design Thinking methodology will continue to be applied throughout program development as the Learning Objectives and the Teaching, Learning, and Assessment strategies are decided. As outlined in Figure 1, a continuous improvement feedback loop is initiated as soon as the proposed course is validated and subsequently deployed. Design Thinking methodology has helped ensure our customers' needs are met by building the curriculum framework around competencies identified by both industry and academia while ensuring the students engage in a significant learning experience through experiential and applied learning using the latest immersive technologies. This will be the first programme using this delivery methodology, but in time it is envisaged further programmes, across all STEM discipline areas, will be developed incorporating this approach.

## 6. Conclusions

This paper described an on-going innovative project in engineering education reform. The authors are unaware of other published work in relation to the use of DT methodology in the co-design and co-development of engineering curricula with industry. A report and set of recommendations for other institutions interested in using the REEdI approach in curriculum design or reform will be produced during the project. The authors aim to have this overall body of work presented internationally as an extended journal article and conference presentation.

**Author Contributions:** Conceptualization, F.B., J.W., D.R. and E.B.; Methodology, F.B., J.W., D.R. and E.B.; Formal analysis, F.B.; investigation, F.B., C.G. and P.K.; resources, J.W.; data curation, F.B., C.G. and P.K.; writing—original draft preparation, F.B., C.G. and P.K.; writing—review and editing, J.W., D.R. and E.B.; visualization, F.B.; supervision, J.W., D.R. and E.B.; project administration, F.B. and J.W; funding acquisition, F.B. and J.W. All authors have read and agreed to the published version of the manuscript.

**Funding:** This research was funded by the Higher Education Authority Human Capital Initiative Pillar 3.

**Institutional Review Board Statement:** Not applicable.

**Informed Consent Statement:** Not applicable.

**Data Availability Statement:** The data presented in this study is available on request from the authors.

**Conflicts of Interest:** The authors declare no conflict of interest.

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
