# Peer review of "REEdI Design Thinking for Developing Engineering Curricula"

_education, doi:10.3390/educsci12030206_

Round 1

Reviewer 1 Report

This paper approaches and interesting and up-to-date topic.

It has a rather descriptive character and this is not a strong point.

Punctual observations for its improvement are:

  • the abstract should emphasize on the results of the present study, not at future plans.
  • statements such as "engineering education ...... a barrier to economic growth" need to be referenced.
  • the authors have to have in mind that the journal addresses an international audience that is not familiar with specific projects/programs that go on in different other countries (such as REEdI program in Ireland)
  • therefore the REEdI program/project needs to be described, what is a ReedI department in an university, what is the REEdI framework
  • acronyms should always be explained first time when appear in the text (VR/AR - virtual reality/augmented reality)
  • when you start talking about your design model, make reference in the text to Fig 1.
  • not clear if the design model presented is entirely proposed by you OR it is taken from an existing methodology
  • the paper needs to make clear that it presents the results for the first step in the proposed model for DT Curriculum Development
  • the methodology of your data collection needs to be presented in more details:
    • how many workshops? when? who were the participants national and/or international
    • justify data collection technique
    • better explain the roles of moderators vs. facilitators in the workshops
    • describe step-by-step how your data collection- workshops were organized. You make reference to what happened during workshops without beforehand presenting the organization of them
  • still editing mistakes need to be addressed (p. 2, raws 54, 73; p. 3 raws 121, 142; p. 4 raws 157; table 3 - six-sigma

Reviewer 2 Report

The authors describe an on-going research of how innovative design thinking methodology can be applied for curriculum development. What makes it unique is the fact that there are no other published works in relation to the research.

The methodology proposed by the authors seems to be perfectly reasonable with all the necessary points covered and discussed. The key strength of this paper is that it is nicely structured and paced. Moreover, the area is relevant. I enjoyed reading through it and learned a lot. What makes it even better is the fact that the authors are taking various feedback into account, adjusting their research whenever needed. The results are well presented and are of high quality. This project will pose a great value for professionals working in the higher education field.

However, some very minor adjustments should be considered to improve the quality of the paper:

  • The conclusion and future work mentioned in “Discussion” section are very brief (lines 355 – 370). That should be expanded to show exactly what are the next steps in the project and how they are going to help with the on-going project development. That will add value to this paper and connect it better with the planned journal article and conference presentation.

Once this is amended, the paper will be suitable for publication in this journal.
